# Dynamic imaging of lithium in solid-state batteries by *operando* electron energy-loss spectroscopy with sparse coding

Yuki Nomura [1✉], Kazuo Yamamoto [2], Mikiya Fujii[1], Tsukasa Hirayama[2,3], Emiko Igaki[1] & Koh Saitoh[3]

Lithium-ion transport in cathodes, anodes, solid electrolytes, and through their interfaces plays a crucial role in the electrochemical performance of solid-state lithium-ion batteries. Direct visualization of the lithium-ion dynamics at the nanoscale provides valuable insight for understanding the fundamental ion behaviour in batteries. Here, we report the dynamic changes of lithium-ion movement in a solid-state battery under charge and discharge reactions by time-resolved *operando* electron energy-loss spectroscopy with scanning transmission electron microscopy. Applying image denoising and super-resolution via sparse coding drastically improves the temporal and spatial resolution of lithium imaging. Dynamic observation reveals that the lithium ions in the lithium cobaltite cathode are complicatedly extracted with diffusion through the lithium cobaltite domain boundaries during charging. Even in the open-circuit state, they move inside the cathode. *Operando* electron energy-loss spectroscopy with sparse coding is a promising combination to visualize the ion dynamics and clarify the fundamentals of solid-state electrochemistry.

[1] Technology Innovation Division, Panasonic Corporation, 3-1-1 Yagumo-naka-machi, Moriguchi, Osaka 570-8501, Japan. [2] Nanostructures Research Laboratory, Japan Fine Ceramics Center, 2-4-1 Mutsuno, Atsuta, Nagoya, Aichi 456-8587, Japan. [3] Institute of Materials and Systems for Sustainability, Nagoya University, Furo-cho, Chikusa, Nagoya, Aichi 464-8603, Japan. ✉email: nomura.yuki001@jp.panasonic.com

Li-ion batteries[1–3] (LIBs) are widely used not only for portable electronic devices, but they have also been used in hybrid/electric vehicles in recent years. Intercalation-type cathodes, such as $LiCoO_2$[4] and $LiFePO_4$[5], are used for practical LIBs because of their advantages in terms of safety, lifetime, energy density, and so forth. In common LIBs, Li ions are released from the cathode materials during the charge reaction, and they are then incorporated in the host structures of the cathode during the discharge reaction. Intercalation and deintercalation of Li ions cause structural changes in the host structures, such as expansion/contraction of the crystal lattice, migration of transition metals, and loss of oxygen, which result in a serious decrease in the electrochemical energy output of LIBs. Therefore, the intercalation and deintercalation processes of Li ions have been intensively investigated to develop high-performance electrodes that can suppress the capacity and voltage decay[6–10].

Intercalation-type cathodes are classified into two types from the viewpoint of the phase transformation during lithiation and delithiation. One type is two-phase transformation cathodes such as $LiFePO_4$, where fully delithiated ($FePO_4$) regions stably form, and the boundary between the delithiated ($FePO_4$) and lithiated ($LiFePO_4$) regions proceeds in a single particle during the reactions. The other type is solid-solution transformation cathodes such as $LiCoO_2$, where all of the regions in the cathode are gradually delithiated and lithiated, and thus no boundaries are formed in a single particle. However, recent studies have shown that a nanoscale and nonequilibrium transformation occurs at the interface via a solid-solution transformation, even in $LiFePO_4$ which is considered to be a two-phase transformation cathode[8,11,12]. Moreover, local variation of the Li-ion concentration has also been reported[13]. The nonequilibrium and inhomogeneous solid-solution transformation is considered to play an important role in the high-rate capability of $LiFePO_4$. Thus, revealing the Li-ion dynamics in the nonequilibrium reaction processes is the key to enhance the electrochemical properties of LIBs. However, much less research has been performed on in situ analyses focusing on the nonequilibrium intercalation processes of solid-solution transformation cathodes such as $LiCoO_2$, and thus it is still unclear how Li ions move in the materials at the nanometre scale.

In situ or *operando* transmission electron microscopy (TEM) is expected to clarify the above questions. Intensive research efforts over the past decade have established techniques to operate battery reactions in transmission electron microscopes[7,14]. Some research groups have used piezo-driven tip TEM holders[7] or liquid cell TEM holders[15] to observe the morphological or crystallographic changes owing to lithiation and delithiation in real-time, for example, lithiation of Si anodes[16] and dendrite growth of Li-metal anodes[15]. However, using these techniques, it is difficult to quantitatively visualize the Li-ion dynamics in intercalation-type active materials such as $LiCoO_2$ because intercalation-type materials do not exhibit large morphological and crystallographic changes during the reactions. One method to observe the Li-ion dynamics in intercalation-type materials is electron energy-loss spectroscopy (EELS), which directly detects Li signals at the nanometre scale. We have used scanning TEM with EELS (STEM-EELS) to quantitatively visualize the changes in the Li distribution at the same area of a $LiCoO_2$ thin film in a solid-state LIB (SSLIB)[17]. However, the temporal resolution of the STEM-EELS measurement was not sufficient to observe the Li-ion dynamics, because a long acquisition time was required to obtain weak Li signals from the EEL spectra, and only four images of the Li distribution were obtained during the charge/discharge reactions in our previous study. To observe the dynamic transformation processes of $LiCoO_2$, the EELS dataset needs to be recorded at a much higher scanning speed to improve the temporal resolution of Li detection. However, the Li maps from EELS at a higher scanning speed are commonly too noisy to display the Li distribution because of the low signal-to-noise ratio (SNR).

In the present study, to drastically enhance the temporal resolution with a sufficient SNR, we use sparse coding (SC)[18–20], which is a machine learning technique for image processing that uses the hidden features of the Li distribution. We demonstrate that SC reconstruction enables the low-SNR Li maps recorded at a higher scanning speed to be denoised and super-resolved. We combine 157 images of the Li distribution as a movie that successfully shows the dynamical behaviour of Li ions in a $LiCoO_2$ thin-film cathode during the charge/discharge reactions. The movie also shows that the Li ions are not monotonically extracted from the cathode in the charged state and the Li ions move inside the cathode even in the open-circuit state of the battery.

## Results

**Configuration of the SSLIB.** A schematic of the SSLIB used in this study is shown in Fig. 1a. A 50-μm-thick $Li_{1+x+y}Al_x(Ti,Ge)_{2-x}Si_yP_{3-y}O_{12}$ (LASGTP) sheet (Ohara Inc., Japan) was used as the solid electrolyte. A 230-nm-thick $LiCoO_2$ film cathode was deposited on the LASGTP sheet by pulsed laser deposition (PLD) at 550 °C. Au and Pt current collectors were then deposited by the common sputtering method. An in situ formed anode fabricated by electrochemical decomposition of LASGTP was used as the anode[21,22]. One part of the cathode side of the battery sample (red dashed rectangle in Fig. 1a) was thinned by a focused ion beam (FIB) for TEM observation. The charge and discharge curves of the thinned SSLIB sample cycled at a constant current of 50 nA (about 0.6 C rate) and subsequent constant voltages of 1.8 and 1.0 V are shown in Fig. 1b. Comparison of the charge and discharge curves of the original and FIB-processed thin film batteries is shown in the Supporting Information of our previous study[17]. The TEM sample stably worked as a LIB with a potential plateau at about 1.6 V, which is consistent with other reported values[23–25]. The configuration of the biasing TEM holder and the method to bias the SSLIB are given in the Supplementary Fig. 1 and Supplementary Methods. An annular dark-field (ADF) STEM image of the sample is shown in Fig. 1c. The $LiCoO_2$ cathode film was deposited without voids, and a sharp interface formed with the LASGTP sheet. The dark gray regions in LASGTP are the grains of $AlPO_4$ that do not have Li ionic conductivity, and the bright gray regions are the grains of $Li_{1+x}Al_xGe_yTi_{2-x-y}P_3O_{12}$ and $Li_{1+x+3z}Al_x(Ge,Ti)_{2-x}(Si_zPO_4)_3$ that have high ionic conductivity of $1 \times 10^{-4}$ S/cm at room temperature[22]. The typical phase distribution and Li concentration in $LiCoO_2$ cathode films investigated in our previous study are shown in Fig. 1d[17]. In the $LiCoO_2$ film deposited by PLD under the same conditions, pillar $LiCoO_2$ grains formed (indicated by black lines in Fig. 1d), and a mixture of $LiCoO_2$ and electrochemically inactive $Co_3O_4$ also formed in the film. $Co_3O_4$ was more concentrated near the $LiCoO_2$/LASGTP interface (blue and green regions, 10–20 nm thick) than at the Au side because some Li ions diffused from $LiCoO_2$ to LASGTP during deposition of $LiCoO_2$ at 550 °C.

**Image denoising and super-resolution.** *Operando* STEM-EELS records four-dimensional (4D) datasets consisting of a time series (1D) of three-dimensional (3D) EEL spectrum images (SIs). In the present study, we used a two-step strategy to enhance the temporal resolution of *operando* STEM-EELS using the hidden features in the 4D dataset. The first step was noise reduction of the EEL spectra in the 4D dataset. This was accomplished by multiple linear least squares (MLLS) fitting of the EEL spectra

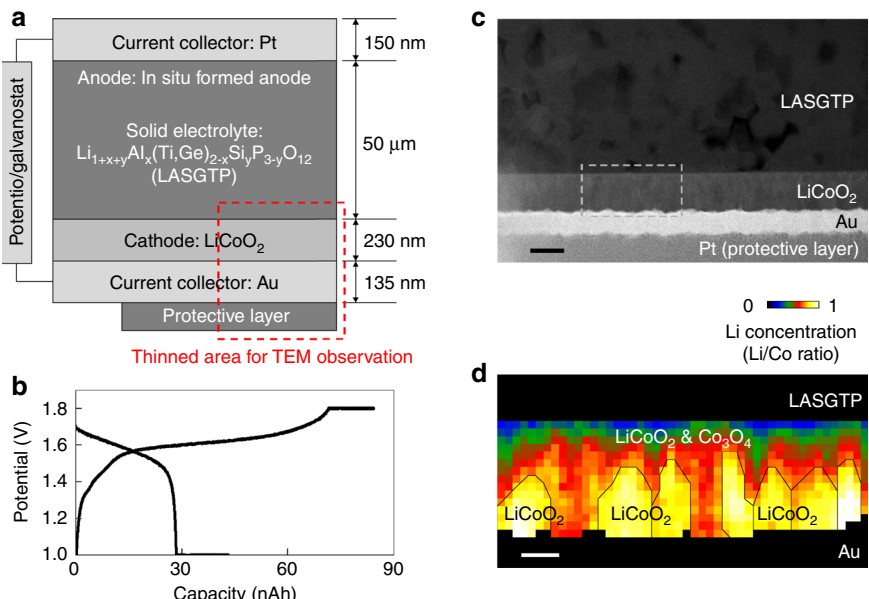

**Fig. 1 Configuration of the SSLIB. a** Schematic of the SSLIB used in this study. The red dashed rectangle shows the thinned area for TEM observation. **b** Charge and discharge curves of the thinned SSLIB sample operated in the transmission electron microscope. **c** Cross-sectional ADF-STEM image around the interface between LiCoO$_2$ and LASGTP. **d** Phases and Li concentration in the LiCoO$_2$ cathode film investigated in our previous study[17]. Reprinted with permission from ref. [17]. Copyright (2018) American Chemical Society. The scale bars in **c** and **d** are 200 and 50 nm, respectively.

using reference EEL spectra[26]. The second step was applying SC for drastic noise reduction and super-resolution using the 2D morphological features of the Li distribution.

To perform the above approach, we acquired the following three datasets in the experiments. First, a 4D dataset consisting of a time series of 157 3D SIs acquired at the low-dose condition in the rectangle region shown in Fig. 1c to visualize the Li-ion dynamics in the LiCoO$_2$ film. The SIs were serially acquired in the same region during the charge and discharge reactions (under the LIB operational condition shown in Fig. 1b, that is, the *operando* condition). Second, 1D reference EEL spectra acquired from reference Li$_x$CoO$_2$ ($x = 0.99$, 0.59, 0.39, and 0.03) particles for MLLS fitting. Third, 2D Li maps as training images for SC to extract the morphological features of the Li distribution. We denoised and super-resolved the 157 sets of the low-dose 3D SIs (the first dataset) using the reference 1D EEL spectra (the second dataset) and the extracted 2D morphological features of the Li distribution (the third dataset).

The reference EEL spectra obtained from electrochemically delithiated Li$_x$CoO$_2$ ($x = 0.99$, 0.59, 0.39, and 0.03) particles (the second dataset) are shown in Fig. 2a–d. Conventional methods used to quantify the atomic ratio using the EEL spectra, such as the three-window method, cannot be applied to LiCoO$_2$ because part of the Li–K edge overlaps with the Co–M edge at about 60 eV, as shown in Fig. 2a–d. To quantify the Li concentration in LiCoO$_2$, we used the $S_A/S_B$ method proposed by Kikkawa et al.[27]. They defined areas of $S_A$ and $S_B$ for the intense peak of the Li–K edge components at around 62 eV and showed that the $S_A/S_B$ ratio is proportional to $x$ in Li$_x$CoO$_2$. Note that the thickness variance of a TEM sample does not have a large effect on the $S_A/S_B$ method because the method measures the Li concentration through the atomic ratio of Li/Co. One of the problems with the $S_A/S_B$ method is the low robustness to noise. This is because the boundary between $S_A$ and $S_B$ is simply determined by the straight line that intersects the EEL spectrum at 60.3 and 62.5 eV. The SIs recorded with high temporal resolution, that is, the low-dose condition, cause low SNR of the EEL spectra and uncertainty of Li quantification by the $S_A/S_B$ method. For example, an ADF-STEM

image simultaneously acquired with one SI in the LiCoO$_2$ cathode film (one of the first datasets) is shown in Fig. 2e. The typical EEL spectrum at the pixel indicated by point A in Fig. 2e is shown by the red line in Fig. 2h that has low SNR, where the STEM probe current, dwell time, and resolution were 260 pA, 0.01 s/pixel, and 58 × 35 pixels, respectively. The corresponding Li concentration (Li/Co ratio) map is shown in Fig. 2f. The Li map was too noisy to identify the pillar LiCoO$_2$ grains shown in Fig. 1d. Note that the Li/Co ratio in the regions of LASGTP and Au was set to be zero because the Li/Co ratio could not be defined in LASGTP and Au. To denoise the EEL spectra, we fitted the noise-free reference spectra shown in Fig. 2a–d to the original low-SNR SI by MLLS fitting in the energy range 60–65 eV (see the yellow region in Fig. 2h). This range was selected to decrease the plural inelastic scattering effects for the Li-K edge, which is different from the higher energy side (ca. 65–80 eV). The blue spectrum in Fig. 2h was extracted from the MLLS-fitted SI. It is clear that the noise component was significantly reduced without losing the features of the Li-K edge around 62 eV. A Li map obtained from this MLLS-fitted SI is shown in Fig. 2g. The pillar LiCoO$_2$ grains became observable because of noise reduction by MLLS fitting.

Next, we applied the SC technique to denoise and super-resolve the Li maps. SC is a patch-based method that extracts the hidden features, the so called "dictionary", from the training images. The details of SC are described in the "Methods" section. We used a third dataset of SIs recorded with a high electron dose and high spatial resolution at static states before and after the charge and discharge reactions. In the static states, the electrochemical reactions did not proceed, and thus the acquisition time was not limited. The high-quality (HQ) Li maps obtained from the high-dose and high-resolution MLLS-fitted SIs were used as the training images to obtain the dictionary. The HQ Li maps obtained from the same area (Fig. 1c) in the 0% charged, 100% charged, and 51% discharged states are shown in Fig. 3a–c. The dwell time and resolution were 0.04 s/pixel and 110 × 45 pixels, respectively. We also acquired low-dose and low-resolution SIs with about 16 times higher temporal resolution from the same area and in the same states as Fig. 3a–c at a dwell time of 0.01 s/pixel and

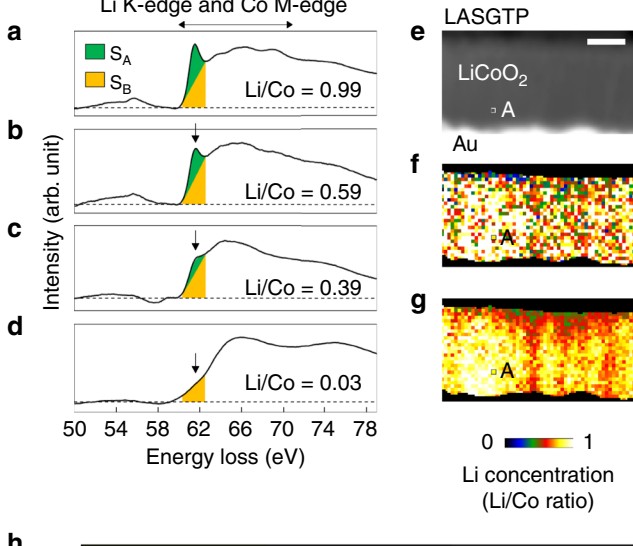

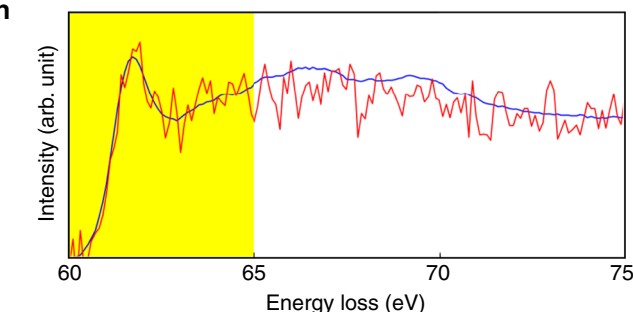

**Fig. 2 Quantification and denoising of the EEL spectra by MLLS fitting.**
**a–d** Reference EEL spectra acquired from delithiated reference $Li_xCoO_2$ particles ($x = 0.99$, 0.59, 0.39, and 0.03) used for the $S_A/S_B$ method and MLLS fitting. Reprinted with permission from ref. [17]. Copyright (2018) American Chemical Society. **e** ADF-STEM image where the SIs were acquired. **f** Li concentration map obtained from the original low-dose SI with low SNR. **g** Li concentration map obtained from the MLLS-fitted SI. The pillar domains of $LiCoO_2$ can be observed. **h** EEL spectra at the pixel indicated by point A in **e–g**. The red and blue spectra were extracted from the original and MLLS-fitted SI, respectively. The scale bar in **e** is 100 nm.

resolution of $55 \times 22$ pixels, which corresponded to about 12 s to take one image. The low-quality (LQ) Li maps obtained from the low-dose MLLS-fitted SIs were used as the test images (Fig. 3d–f). The LQ Li maps (Fig. 3d–f) were very noisy because each dose and resolution was a quarter of those used to obtain the HQ Li maps (Fig. 3a–c). SC was used to extract the features of the Li distribution (dictionary) from the HQ Li maps. The dictionary, in which the number and size of the dictionary were 5 and $18 \times 18$ pixels, respectively, optimized using the cross-validation method is shown in Fig. 3g. The details are described in the "Methods" section. The LQ Li maps were denoised and super-resolved by SC with the dictionary. The SC-reconstructed images from the LQ Li maps are shown in Fig. 3h–j. The reconstructed Li maps excellently reproduced the HQ Li maps of Fig. 3a–c, except for the noise. The residuals between the HQ (Fig. 3a–c) and SC-reconstructed (Fig. 3h–j) Li maps are shown in Fig. 3k–m, indicating that the random noise was effectively removed by SC. Comparison of reconstructed images processed with different parameters (the number and size of the dictionary) is shown in the Supplementary Fig. 2 and Supplementary Note 1. We calculated the peak SNR (PSNR) to quantitatively evaluate reproduction of the images by SC, where a higher PSNR value indicates higher reproduction by SC. The PSNRs of the original LQ Li maps, SC-reconstructed Li maps, and bilinear interpolated

LQ Li maps for comparison are given in Table 1. The PSNR values of the SC-reconstructed Li maps were higher than those of the original LQ and bilinear interpolated Li maps in all of the states. Therefore, SC reconstruction was effective to retrieve the true Li maps from low-dose SIs recorded at a higher scanning speed.

**Dynamic imaging of Li concentration.** Part of the time series of 157 ADF-STEM images and the corresponding Li maps from the original low-dose SIs, MLLS-fitted SIs, and SC reconstruction are shown in Fig. 4a–d, respectively. Movies of the whole time series of Fig. 4a, b, d are provided in the Supplementary Movie 1, in which the nanoscale Li dynamics during the charge/discharge reactions were clearly observed for the first time. In the snapshots shown in Fig. 4, the charge/discharge capacity, percentage of the state of charge (SOC), and image number of the series are indicated at the left-hand side of each ADF-STEM image. The random noise in the original Li maps (Fig. 4b) was removed by the MLLS fitting process, and the Li concentration in each $LiCoO_2$ grain was retrieved (Fig. 4c). In addition, as shown in Fig. 4d, the quality of the Li maps was drastically improved by denoising and super-resolving in the SC-reconstruction processes. Extraction and insertion of Li ions during the charge and discharge reactions were also clearly observed. The changes in the average Li concentrations in area 1 (large), 2 (middle), and 3 (small) indicated in Fig. 4b–d are shown in Fig. 4e–g, respectively, where the horizontal axis is the battery capacity in the charged and discharged states. The Li concentrations of the original (black) and MLLS-fitted (blue) SIs were calculated from the average spectra in the areas. Averaging the Li signals in each area improved the SNR of the EELS spectra, and it is therefore convenient to more quantitatively observe the rates of the Li extraction/insertion reactions in each area. The plots of the MLLS-fitted and SC-reconstructed Li maps showed significant suppression of the noise without artefacts. As the capacity increased in the charged state, the Li ions were extracted from the $LiCoO_2$ cathode, and, as a result, the Li concentration in each area decreased by a similar rate. In the discharged state, the Li concentration in each area increased by the insertion reaction, but the amount of the Li concentration change in the discharged state was lower than that in the charged state. This irreversible capacity is because of in situ formation of anode active materials, where some Li ions were trapped in the anode side of LASGTP crystals[21,22,24,25]. The total dose through the present *operando* STEM-EELS was about $4.3 \times 10^5$ [electron/Å²], which is the same order of magnitude as our previous *operando* STEM-EELS (about $1.7 \times 10^5$ [electron/Å²])[17] and typical atomic-resolution STEM for $LiCoO_2$ (about $2.8 \times 10^5$ [electron/Å²])[28]. Thus, we considered that electron irradiation did not have a large effect on the results. The detailed description of the electron beam effect is given in the Supplementary Fig. 3 and Supplementary Note 2.

**Spatial variation of the Li concentrations.** From the Li-ion dynamics, we observed spatial variation of the Li concentrations in the $LiCoO_2$ cathode during the charge and discharge processes. An ADF-STEM image of the cathode and the change in the average Li concentration in the entire cathode film as a function of capacity are shown in Fig. 5a, b, respectively. The average Li concentration was calculated as the average value of the SC-reconstructed Li maps in the entire cathode film. The average Li concentration almost proportionally decreased and increased as a function of the charge and discharge capacity, respectively. During the open-circuit state for 30 min between the charge and discharge states (purple region in Fig. 5b), the average Li concentration in the entire cathode film did not change.

The changes in the Li concentration at local points 1–3 and 4–6 indicated in Fig. 5a are shown in Fig. 5c, d, respectively, in which

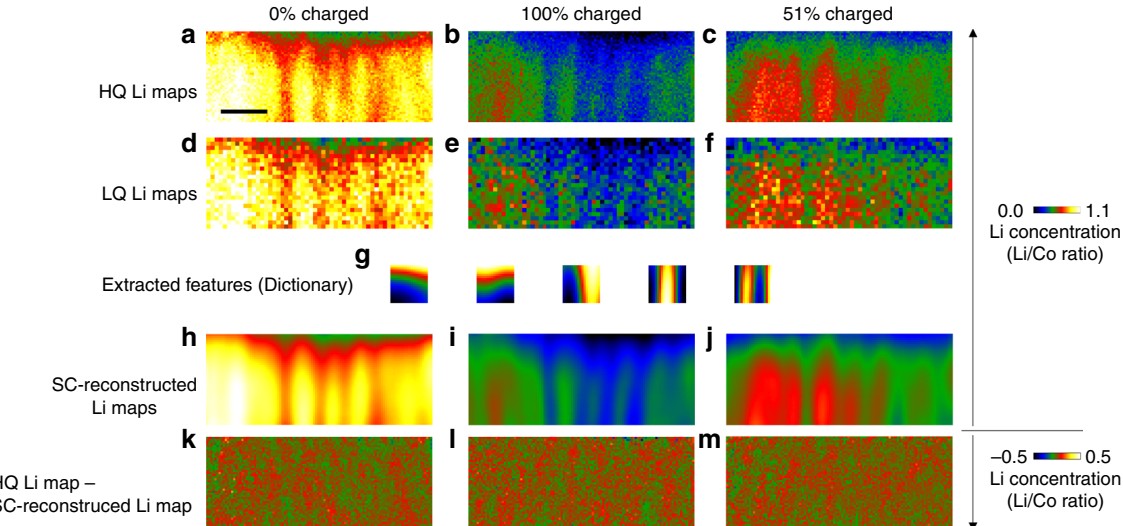

**Fig. 3 Denoising and super-resolution of the Li maps by SC. a–c** High-quality (HQ) Li maps of the cathode film in the 0% charged, 100% charged, and 51% discharged states, respectively. These images were acquired from the SIs with a high dwell time of 0.04 s/pixel and high resolution of 110 × 45 pixels. **d–f** Low-quality (LQ) Li maps of the same region and states as **a–c**. The dwell time and resolution were a quarter of those used for the HQ Li maps (**a–c**). **g** Extracted morphological features (the so-called dictionary) of the Li distribution. **h–j** Denoised and super-resolved Li maps obtained from the LQ Li maps (**d–f**) by SC. **k–m** Residuals between the HQ (**a–c**) and SC-reconstructed (**h–j**) Li maps. Note that the HQ Li maps obtained in the same state as the test images were not used as the training images to avoid overfitting, that is, only the HQ Li maps in the 100% charged and 51% discharged states (Fig. 3b, c) were used as the training images for reconstruction of the LQ Li map in the 0% charged state (Fig. 3d). The scale bar in **a** is 100 nm.

**Table 1 PSNR (dB) values of the original LQ, SC-reconstructed, and interpolated (bilinear interpolation) LQ Li maps.**

|  | 0% charged | 100% charged | 51% discharged |
|---|---|---|---|
| LQ Li maps | 20.5 | 19.6 | 20.6 |
| SC-reconstructed Li maps | **25.9** | **25.1** | **26.5** |
| bilinear interpolated LQ Li maps | 23.5 | 22.3 | 23.3 |

A higher PSNR indicates that the reconstructed Li map is more similar to the standard image. Because we could not obtain noise-free Li-ion maps in the practical experiments, we used the HQ Li maps as the standard images.
Bold values show the highest PSNR at each state.

the concentration values were extracted from the SC-reconstructed Li maps. Note that the cathode film was composed of pillar-structured domains of LiCoO$_2$, as shown in Fig. 1d. Thus, the three points 1–3 were chosen in one domain, and the other points 4–6 were chosen in the other domain. During stage A in the charged state from 3 to 18 nAh (yellow region in Fig. 5b–d), the Li concentrations at points 1–3 decreased (Fig. 5c), where the concentration at point 1 decreased faster than those at points 2 and 3 because point 1 was closer to the LASGTP solid electrolyte. The Li concentrations at points 1–3 then slightly increased in stage B from 39 to 53 nAh (pink region in Fig. 5b–d). This means that the Li ions were not monotonically extracted in the pillar domain during the charging process. Because the average concentration in the entire LiCoO$_2$ proportionally decreased as a function of charge capacity, unintended Li-ion reversal locally occurred in the cathode film during the reaction. In the open-circuit state for the 30 min between charge and discharge states, the Li concentrations at points 1–3 decreased (Fig. 5c), although the Li ions were not extracted from the cathode. In contrast, at points 4–6 in the other domain, the Li

concentrations increased in stage A, decreased in stage B, and increased in the open-circuit state, which are the opposite changes to those at points 1–3. It seems that Li-ion extraction and Li-ion diffusion between the domains simultaneously occurred in the cathode film. In the discharged state, such complicated behaviour did not occur. The Li concentrations at points 1–6 proportionally increased with almost the same rate (Fig. 5c, d). Therefore, these dynamic changes of the Li concentration indicate that the local movements of the Li ions in the cathode film were different for the charge and discharge states. To clarify the reason for this behaviour, further experiments are required.

We directly observed the above spatial variation in the Li maps. Some snapshots of the movie showing the changes in the Li maps at stages A, B, and C are shown in Fig. 5e–j. At the start of stage A (3 nAh, Fig. 5e), the Li concentration around points 1–3 in the left domain was higher than that around points 4–6 in the right domain. However, at the end of stage A (18 nAh, Fig. 5f), the Li concentration in the left domain was as high as that in the right domain. This can be interpreted as the Li ions moved through the boundaries of neighbouring pillars parallel to the cathode/solid-electrolyte interface. The same phenomena were observed in stages B and C. At the start of stage B (39 nAh, Fig. 5g), the Li concentration around points 1–3 in the left domain was slightly higher than that around points 4–6 in the right domain. However, at the end of stage B (53 nAh, Fig. 5h), the Li concentration in the left domain was much higher than that in the right domain. The concentration change in the open-circuit state for 30 min between charging and discharging was the most evident. Before the open-circuit state (Fig. 5i), the Li concentration around points 1–3 in the left domain was much higher than that around points 4–6 in the right domain. However, after the open-circuit state (Fig. 5j), the Li concentration in the left domain was as high as that in the right domain. Therefore, we considered that the Li ions moved parallel as well as vertical to the interface during the charge reaction. Moreover, the Li ions moved inside the cathode even if the macroscopic battery reactions stopped in the open-circuit state.

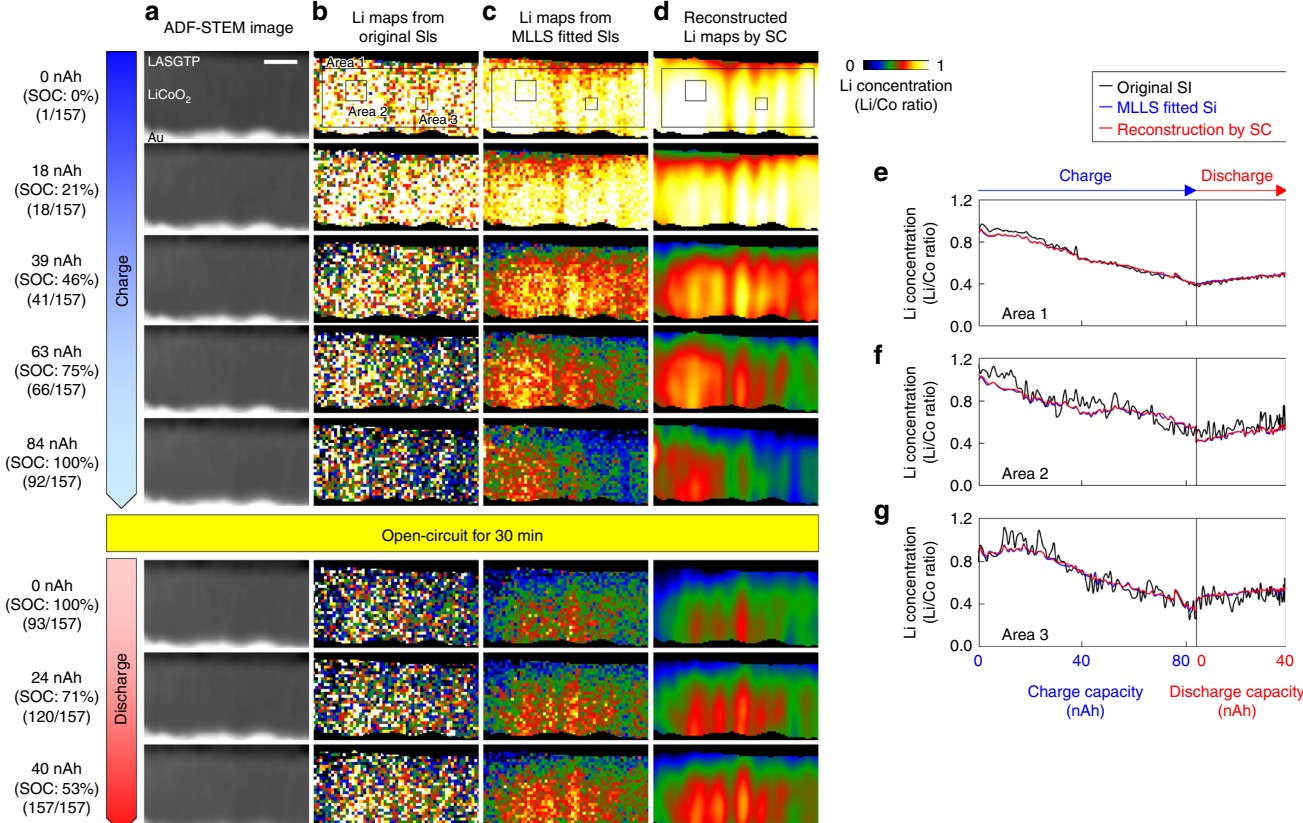

**Fig. 4 Comparison of the original and denoised Li maps during the electrochemical reactions. a–d** Part of the time series of 157 **a** ADF-STEM images, **b** Li maps obtained from the original SI, **c** Li maps obtained from the MLLS-fitted SI, and **d** Li maps obtained by SC reconstruction. **e–g** Changes in the average Li concentrations in the areas indicated in **b–d**. The results indicated that MLLS fitting and SC reconstruction efficiently denoised and super-resolved without producing serious artefacts. The scale bar in **a** is 100 nm.

## Discussion

The non-monotonic extraction shown in Fig. 5c, d is a surprising phenomenon in battery research fields. However, we consider that the phenomenon was not caused by characterization error but by the lateral diffusion of Li ions because of the following experimental evidence. One of the reasons is that the average Li concentration in the whole LiCoO₂ film monotonically decreased, as shown in Fig. 5b, which is consistent with galvanostatic charging. This shows that our STEM-EELS correctly measured at least the average Li-ion concentration. Moreover, non-monotonic extraction was suggested in the raw STEM-EELS data without spectrum fitting and SC, as shown in the black plots in Fig. 4f, g. The black plots were calculated from the unprocessed (original) EELS spectra with only spatial averaging. The plots showed noisy but clear non-monotonic extraction, where the changes in the Li concentration were similar to those in Fig. 5c, d. Therefore, we concluded that the Li ions moved in the lateral direction between the LiCoO₂ domains and were non-monotonically extracted in nanoscale local regions. Recently, Zhang et al.[13] reported nanoscale reversal of the Li concentration in LiFePO₄ nanoparticles by high-resolution TEM (HRTEM) with geometric phase analysis (GPA). They detected the variation of the lattice constant of LiFePO₄ to estimate the Li concentration. They suggested that the reversal of the Li concentration was caused by the variation of the free energy function, which may originate from lattice defects in LiFePO₄. It is well known that the LiCoO₂ film cathode deposited by PLD contains oxygen defects depending on the substrate temperature, oxygen partial pressure, and other parameters. Thus, it is possible that the cathode film contains many lattice defects in the LiCoO₂ domains and at the interfaces between the LiCoO₂ and Co₃O₄ domains. The present study reveals that nanoscale reversal of the Li concentration also occurs in the thin film of the LiCoO₂ cathode. The advantages of the proposed method are the direct detection of Li by EELS and the applicability to polycrystalline materials with a wide field of view, while in situ HRTEM with GPA requires, in principle, atomic-resolution images and single-crystal materials, which are not convenient for analysis of practical battery electrodes.

In conclusion, we have demonstrated that *operando* STEM-EELS with SC enables direct visualization of the Li-ion dynamics in SSLIBs during the charge and discharge reactions. Dynamic observation revealed that the Li ions not only moved in the vertical direction to the electrode/solid-electrolyte interface, but they also moved in the parallel direction, and thus the Li concentration spatially varied at the nanometre scale during the electrochemical reactions. The present study also showed that Li-ion diffusion occurred inside the cathode film even in the open-circuit state. Therefore, the combination of STEM-EELS and SC is a promising *operando* technique for probing the fundamental properties of solid-state electrochemistry.

## Methods

**Preparation of the SSLIB**. The details of PLD deposition, the in situ formed anode, and impedance spectroscopy of the SSLIB are described in the Supporting Information of our previous study[17].

**Preparation of the reference Li$_x$CoO₂ particles**. We prepared Li$_x$CoO₂ particles with different Li-ion concentrations by electrochemical delithiation of

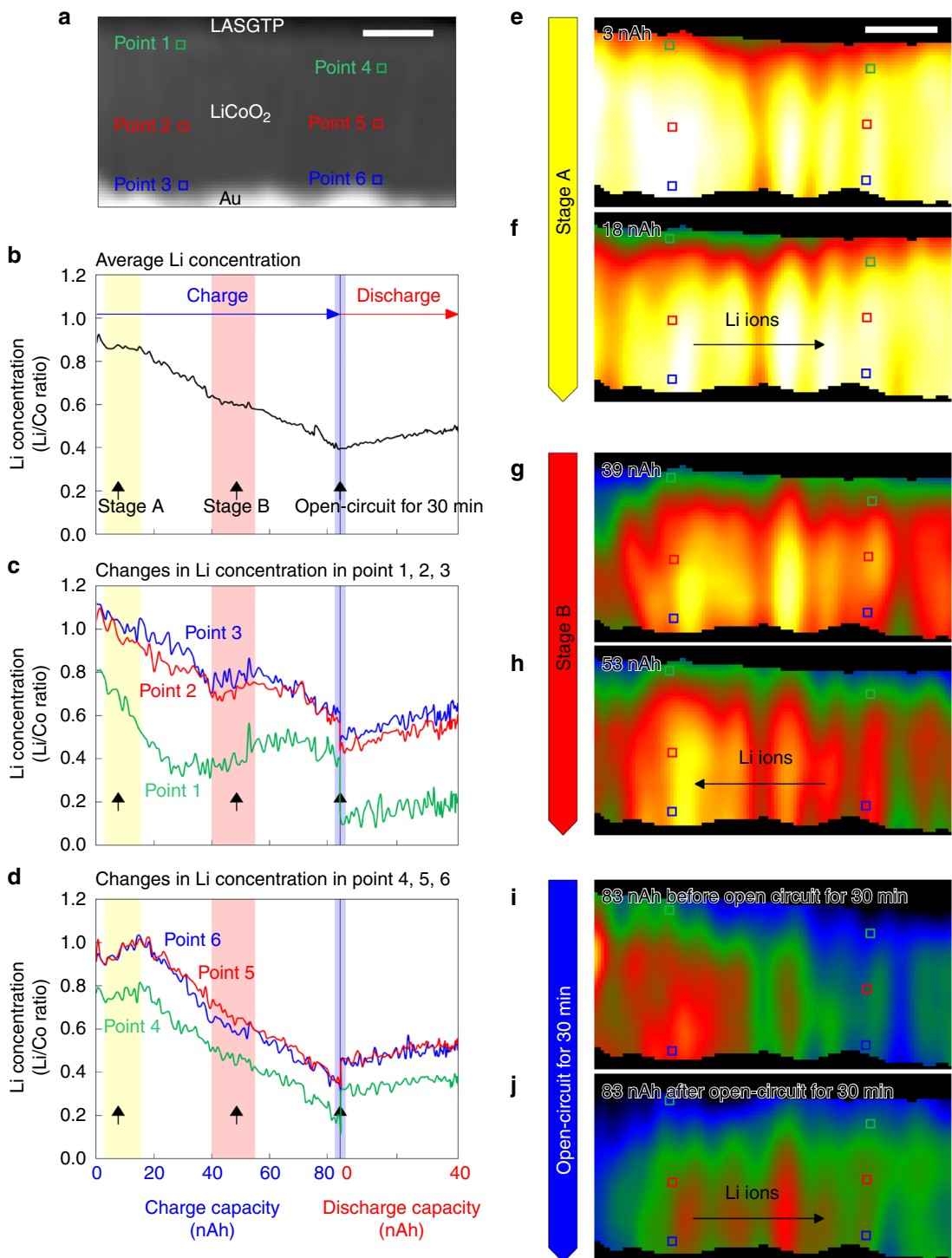

**Fig. 5 Changes in the Li concentration at the nanometre scale. a** ADF-STEM image where the SIs were acquired. **b** Change in the average Li concentration in the entire cathode film in **a**. **c**, **d** Changes in the Li concentrations at points 1–3 and 4–6, respectively. **e–j** Changes in the Li maps at stages A (3–18 nAh), B (39–53 nAh), and C (open-circuit state for 30 min between the charge and discharge reactions), respectively. The dynamic observation revealed that the Li ions moved not only in the vertical direction to the interface, but also in the parallel direction. The scale bars in **a** and **e** are 100 nm.

commercially available LiCoO$_2$. The details are also described in the Supporting Information of our previous study[17].

**Acquisition of the EEL spectra.** The EEL spectra were obtained with a 200 kV electron microscope (ARM-200F, JEOL Ltd.) equipped with a cold-field emission gun and a spectrometer (Gatan imaging filter Quantum ER, Gatan Inc.). The energy dispersion and full width at half maximum of the zero-loss peak were 0.05 eV/pixel and 0.35–0.45 eV, respectively. To minimize the anisotropy effect, the collection semi-angle was set to a relatively large angle of about 88.9 mrad.

**SC**. SC is a patch-based method to find the hidden features (dictionary) in training images (high-dose and high-resolution Li maps)[29]. In SC, patches of $a \times a$ pixels are extracted from the high-resolution training image of $W \times H$ pixels ($W, H > a$), where the extraction region is shifted by one pixel in a raster scan order from left to right and top to bottom. The $W \times H$ training image has $N = (W - a + 1) \times (H - a + 1)$ overlapping patches. Each patch is flattened to a one-dimensional vector and stored in a high-resolution data matrix: $\mathbf{y}_{HR,i} \in R^M$ ($i = 1 - N$), where $M$ is the number of pixels in the patches ($= a^2$). SC finds a representation of data matrix $\mathbf{y}_{HR,i}$ as a product of the high-resolution bases matrix $\mathbf{D}_{HR} \in R^{M \times n}$ (dictionary) and corresponding

sparse weights matrix $\mathbf{C}_{\mathrm{HR},i} \in R^n$ such that

$$\mathbf{y}_{\mathrm{HR},i} = \mathbf{D}_{\mathrm{HR}}\mathbf{C}_{\mathrm{HR},i} + \boldsymbol{\varepsilon}_i \quad \text{for all } i, \tag{1}$$

where $n$ is the number of bases and $\boldsymbol{\varepsilon}_i \in R^M$ is noise. The matrices $\mathbf{D}_{\mathrm{HR}}^*$ and $\mathbf{C}_{\mathrm{HR},i}^*$ are learned by minimization of the cost function:

$$\left(\mathbf{D}_{\mathrm{HR}}^*, \mathbf{C}_{\mathrm{HR},i}^*\right) = \mathrm{argmin}_{\mathbf{D}_{\mathrm{HR}}, \mathbf{C}_{\mathrm{HR},i}} \left\{ \left\| \mathbf{y}_{\mathrm{HR},i} - \mathbf{D}_{\mathrm{HR}}\mathbf{C}_{\mathrm{HR},i} \right\|_2^2 + \lambda \left\| \mathbf{C}_{\mathrm{HR},i} \right\|_1 \right\} \quad \text{for all } i, \tag{2}$$

where $\lambda$ is a parameter to control the sparsity of the weights matrix. The first term maintains data fidelity, and the second term enforces sparsity. $a$, $n$, and $\lambda$ are user-chosen hyper-parameters. In the same way as the high-resolution training images, patches of $\frac{a}{d} \times \frac{a}{d}$ pixels are extracted from low-resolution test images of $W' \times H'$ pixels ($W', H' > a/d$), where $d$ is the sampling rate ($d = 2$ in this study). The $W' \times H'$ training image has $N' = \left(W' - \frac{a}{d} + 1\right) \times \left(H' - \frac{a}{d} + 1\right)$ overlapping patches. Each patch is flattened and stored in a low-resolution data matrix: $\mathbf{y}_{\mathrm{LR},j} \in R^K$ ($j = 1 - N'$), where $K$ is the number of pixels in the patches ($=a^2/d^2$). We then define the low-resolution bases matrix $\mathbf{D}_{\mathrm{LR},j}^* \in R^{K \times n}$ using the learned high-resolution bases matrix $\mathbf{D}_{\mathrm{HR}}^*$ and the down sampling matrix $\mathbf{S}_j \in R^{K \times M}$.

$$\mathbf{D}_{\mathrm{LR},j}^* = \mathbf{S}_j\mathbf{D}_{\mathrm{HR}}^* \quad \text{for all } j. \tag{3}$$

The low-resolution bases matrix $\mathbf{D}_{\mathrm{LR},j}^*$ is then applied to the low-resolution data matrix $\mathbf{y}_{\mathrm{LR},j}$ extracted from the test image. The sparse weights matrix $\mathbf{C}_{\mathrm{LR},j}^{**} \in R^n$ are learned by minimization of the cost function:

$$\mathbf{C}_{\mathrm{LR},j}^{**} = \mathrm{argmin}_{\mathbf{C}_{\mathrm{LR},j}} \left\{ \left\| \mathbf{y}_{\mathrm{LR},j} - \mathbf{D}_{\mathrm{LR},j}^*\mathbf{C}_{\mathrm{LR},j} \right\|_2^2 + \lambda \left\| \mathbf{C}_{\mathrm{LR},j} \right\|_1 \right\} \quad \text{for all } j. \tag{4}$$

The super-resolved and denoised low-resolution data matrix $\mathbf{y}_{\mathrm{HR},j}' \in R^M$ is represented as the product of the sparse weights matrix $\mathbf{C}_{\mathrm{LR},j}^{**}$ learned by Eq. (4) and the high-resolution bases matrix $\mathbf{D}_{\mathrm{HR}}^*$ learned by Eq. (2):

$$\mathbf{y}_{\mathrm{HR},j}' = \mathbf{D}_{\mathrm{HR}}^*\mathbf{C}_{\mathrm{LR},j}^{**} \quad \text{for all } j. \tag{5}$$

The super-resolved and denoised data matrix $\mathbf{y}_{\mathrm{HR},j}'$ are converted to two-dimensional patches and combined to form the recovered image, where the intensity of each pixel in the recovered image is the average intensity of each overlapping patch.

For better super-resolution and denoising, it is important to use appropriate hyper-parameters $a$, $n$, and $\lambda$. In the present study, we used the cross-validation method to optimize the hyper-parameters. We calculated 4320 sets of hyper-parameters, where $a = 4, 5, \ldots, 38, 39$, $n = 1, 2, \ldots, 14, 15$, and $\lambda = 10^{-8}, 10^{-7}, \ldots, 1, 10$. The optimized parameters were $a = 18$, $n = 5$, and $\lambda = 1\mathrm{E}{-}2$.

## Data availability
The data that support the findings of this study are available from the corresponding author upon request.

## Code availability
The codes associated with the findings of this study are available from the corresponding authors upon request.

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

## Acknowledgements
We thank Mr. Yukihiro Umetani of Panasonic Corporation for valuable suggestions for the SC program written in Python. We also thank Ms. Mayumi Ohkawa and Mr. Nobuhiko Hojo of Panasonic Corporation for preparing electrochemically delithiated $Li_xCoO_2$ particles. This work was partly supported by a Grant-in-Aid for Scientific Research KAKENHI (JP 17H02792) from the Japan Society for the Promotion of Science. We thank Tim Cooper, PhD, from Edanz Group (www.edanzediting.com/ac) for editing a draft of this manuscript.

## Author contributions
Y.N. performed the STEM-EELS experiments and SC analysis. Y.N. wrote the draft manuscript. K.Y., M.F., T.H., and K.S. revised the manuscript. M.F. discussed SC. E.I. contributed to the discussion and suggestions. All of the authors contributed to discussion of the results and read and commented on the manuscript.

## Competing interests
The authors declare no competing interests.
