## [Peer Review File · Nature Communications]

Reviewers' comments:

Reviewer #1 (Remarks to the Author):

The authors demonstrated in-situ EELS mapping analyses of the LiCoO_2 cathode in a solid-state LIB under operational conditions by applying image denoising and super-resolution with spectrum fitting and sparse coding. The Li-ion dynamics in LIBs during the charge and discharge reactions were visualized with nanometer spatial resolution. I think the method used in this paper is very promising for visualizing the ion dynamics in solid-state electrochemistry and dynamical valence variation in other fields. In my opinion the study might be publishable for your journal.

Reviewer #2 (Remarks to the Author):

Y. Nomura et al. reported Li-ion movement in solid-state Li-ion batteries using time-resolved STEM-EELS techniques. They applied sparse coding, a machine learning technique for image processing, to achieve the improved temporal and spatial resolution of Li imaging. They found that unique information Li ions diffusion in the film, for example, through the LiCoO_2 domain boundaries. It is well known that it is a great challenge to obtain strong signal of Li ions from EELS at fast recording speed, mostly due to the low signal-to-noise ratio (SNR). Their combined STEM-EELS with sparse coding demonstrated significant advances. I support publication of this work in Nature Communications after the authors address the following concerns.

- 1). What is the threshold of electron beam dose to induce Li ion movement? How to quantify the electron beam effects under the measured conditions? This is critical information for understanding the observed Li ion diffusion; to eliminate the electron beam effects.
- 2). The sparse coding technique for image processing has been reported elsewhere. The authors should comment on their approach compared to others.
- 3). It is hard to make judgement on the reliability of the reconstruction images. It seems too good. Since sparse coding image processing is the key of this paper, I think it is necessary to provide comparison of images processed with systematically modified parameters.

Reviewer #3 (Remarks to the Author):

In the present work, the authors visualized Li ions of a solid-state battery in real time by operando EELS. With sparse coding reconstruction, EELS mapping can be clarified, and the resolution is good enough to distinguish different LiCoO_2 domains. This work brings a brand new and useful technique to probe the dynamic processes of lithium ion batteries (LIB). The authors directly visualized an interesting phenomenon: Li ions migrated in the cathode by concentration gradient between different LiCoO_2 domains when the microbattery was in the open-circuit state. This is meaningful to study the LIB microscale mechanism. However, there are still some minor concerns and ambiguities in the present form of the manuscript.

1. The thickness of in situ formed anode in previous research (Journal of Power Sources 266 (2014) 414-421) was 400-700 nm, but the thickness of LASGTP in FIB sample (Fig. 1c) seems to be too thin to keep the character of solid electrolyte after the in situ transformation. Hence, it's important to

prove the LASGTP and anode are still working after the in situ biasing operation. It's better to show the in situ biasing charge-discharge curve to clarify this point.

2. It is necessary to describe the TEM holder and the way to bias the microbattery (STM holder or MEMS chip holder). The previous report had no such information (DOI: 10.1021/acs.nanolett.8b02587). The specific experimental conditions are essential to understand this operando biasing TEM studies.

3. FIB may cause the thickness variance for different parts of the LiCoO₂ cathode especially at the vicinity of boundaries. The influence of thickness on EELS is well-known (DOI: 10.1088/0034-4885/72/1/016502). Does it have an influence on the EELS data, especially the sparse coding reconstruction process?

4. On page 14, the calculation method of the average Li concentration is better to be described in detail at the supporting information.

5. The reason why Li ions were non-monotonic extracted during the charging process should be reasonably discussed (Stage B) since this is very unusual to battery research. This abnormal phenomenon was caused by characterization error or lateral diffusion of Li ions?

Reply to Reviewers

Title: Dynamic imaging of lithium in solid-state batteries by *operando* electron energy-loss spectroscopy with sparse coding

Authors: Yuki Nomura, Kazuo Yamamoto, Mikiya Fujii, Tsukasa Hirayama, Emiko Igaki & Koh Saitoh

We would like to thank the editor and reviewers for reviewing our manuscript and giving constructive feedback. The comments from the reviewers have helped us to substantially improve our manuscript. We have revised the manuscript according to the reviewers' comments. The revisions are summarized as follows:

- 1) We have added the configuration of the TEM holder and the method to bias the micro-battery to the Supplementary Information
- 2) We have added the detailed description of our sparse coding to the Supplementary Information.
- 3) We have added the detailed description of the electron beam effect to the Supplementary Information.

The following are our point-by-point responses to the reviewers' comments. Please read along with a marked copy of the revised manuscript. The reviewers' comments are in boxes and our responses to their comments are shown in blue.

1. Response to Reviewer #1

Comment: The authors demonstrated in-situ EELS mapping analyses of the LiCoO₂ cathode in a solid-state LIB under operational conditions by applying image denoising and super-resolution with spectrum fitting and sparse coding. The Li-ion dynamics in LIBs during the charge and discharge reactions were visualized with nanometer spatial resolution. I think the method used in this paper is very promising for visualizing the ion dynamics in solid-state electrochemistry and dynamical valence variation in other fields. In my opinion the study might be publishable for your journal.

Response: We thank the reviewer for understanding the significance of this work.

2. Response to Reviewer #2

Comment: Y. Nomura et al. reported Li-ion movement in solid-state Li-ion batteries using time-resolved STEM-EELS techniques. They applied sparse coding, a machine learning technique for image processing, to achieve the improved temporal and spatial resolution of Li imaging. They found that unique information Li ions diffusion in the film, for example, through the LiCoO₂ domain boundaries. It is well known that it is a great challenge to obtain strong signal of Li ions from EELS at fast recording speed, mostly due to the low signal-to-noise ratio (SNR). Their combined STEM-EELS with sparse coding demonstrated significant advances. I support publication of this work in Nature Communications after the authors address the following concerns.

Response: We thank the reviewer for giving a positive comment and understanding the significance of our

work.

Comment #1: What is the threshold of electron beam dose to induce Li ion movement? How to quantify the electron beam effects under the measured conditions? This is critical information for understanding the observed Li ion diffusion; to eliminate the electron beam effects.

Response: We consider that the electron beam effect can be categorized into two types for observing and evaluating battery materials using TEM. The first type is deterioration of the crystal structure by atom displacement, e-beam sputtering, e-beam heating, electrostatic charging, and radiolysis (DOI: 10.1016/j.micron.2004.02.003). In LiCoO₂, it is well known that crystal deterioration first appears as “cation mixing” between Li and Co sites. The relationship between the electron dose and cation mixing has been reported by Shim *et al.* (DOI: 10.1021/acsami.9b15608). From their STEM imaging conditions, the calculated threshold dose for cation mixing in LiCoO₂ was more than 2.2×10^7 [electron/Å²]. However, the total dose in our study was 4.3×10^5 [electron/Å²], which is two orders of magnitude lower than the threshold value. Thus, we believe that deterioration of the crystal structure hardly occurred in our experiment. The second type is Li-ion diffusion induced by electric charging of the TEM sample. For example, if the illuminated area is positively charged, Li ions with positive charge might move away from the area. To clarify this effect, we compared two Li-K EEL spectra acquired using weak and strong e-beams (different dose rates), where we assumed that the extent of the electric charging depended on the dose rate. If the second type is effective, the Li-K intensity changes depending on the dose rate. Spectrum (a) in Fig. S3 was acquired using a 0.26 pA probe with 100 s exposure. Spectrum (b) was acquired using a 272 pA probe with 0.1 s exposure, which is almost the same dose rate as our *operando* STEM-EELS. For both of the spectra, the STEM probes were positioned at the same LiCoO₂ region with the same probe size (about 5 nm). The results showed that the Li-K intensities were almost the same. Thus, we concluded that electric charging did not occur and affect Li-ion movement under our STEM-EELS conditions because of the high electron conductivity of LiCoO₂. We have added the above discussion to the Supplementary Information.

Figure S3. Comparison of two Li-K EEL spectra acquired using (a) 0.26 and (b) 272 pA e-beams. The STEM probes were positioned at the same LiCoO₂ region with the same probe size (about 5 nm).

Comment #2: The sparse coding technique for image processing has been reported elsewhere. The authors should comment on their approach compared to others.

Response: We thank the reviewer for the suggestion. As the reviewer mentioned, sparse coding techniques have been applied to electron microscopy images. We have added the explanation of our sparse coding compared with previous studies (refs. 19 and 20) to the Supplementary Information. The most important difference in our method is optimization of the hyper-parameters. In previous studies, the reliability of the reconstructed images was not guaranteed because the hyper-parameters (size of the bases, number of bases, and sparsity) were not optimized. We optimized the hyper-parameters using training images (Fig. 3a–c, high quality Li maps) and the cross-validation method, which estimates the hyper-parameters to provide the minimum error. The quality of the reconstructed image significantly depended on the hyper-parameters, as shown in the response to comment #3. Therefore, we believe that highly quantitative and reliable Li maps were obtained. Comparison of the images processed with different hyper-parameters (size and number of bases) is provided in the Supplementary Information.

Comment #3: It is hard to make judgement on the reliability of the reconstruction images. It seems too good. Since sparse coding image processing is the key of this paper, I think it is necessary to provide comparison of images processed with systematically modified parameters.

Response: We thank to the reviewer for pointing out this issue. As recommended by the reviewer, we have added the comparison of the images processed with different hyper-parameters (size and number of bases) to the Supplementary Information. When smaller size and larger number of bases were used as hyper-parameters (e.g., size of 8 pixels, number of bases of 20, top-right purple rectangle), the noise could not be removed. Conversely, when larger size and smaller number of bases were used (e.g., size of 28 pixels, number of bases of 2, bottom-left black rectangle), the noise was successfully removed, but the contrast of the LiCoO₂ domain disappeared. Using the optimized hyper-parameters (size of 18 pixels, number of bases of 5, centre blue rectangle), efficient noise reduction and preservation of the LiCoO₂ domain contrast were simultaneously achieved. The results showed that hyper-parameter tuning using the training images is important for image processing with sparse coding.

Figure S2. Comparison of images processed with different hyper-parameters (size and number of bases) in sparse coding. The optimized hyper-parameters are 18 pixels and 5 bases (blue rectangle).

3. Response to Reviewer #3

Comment: In the present work, the authors visualized Li ions of a solid-state battery in real time by operando EELS. With sparse coding reconstruction, EELS mapping can be clarified, and the resolution is good enough to distinguish different LiCoO₂ domains. This work brings a brand new and useful technique to probe the dynamic processes of lithium ion batteries (LIB). The authors directly visualized an interesting phenomenon: Li ions migrated in the cathode by concentration gradient between different LiCoO₂ domains when the microbattery was in the open-circuit state. This is meaningful to study the LIB microscale mechanism. However, there are still some minor concerns and ambiguities in the present form of the manuscript.

Response: We thank the reviewer for understanding the importance of this work and the valuable comments.

Comment 1: The thickness of in situ formed anode in previous research (Journal of Power Sources 266 (2014) 414-421) was 400-700 nm, but the thickness of LASGTP in FIB sample (Fig. 1c) seems to be too thin to keep the character of solid electrolyte after the in situ transformation. Hence, it's important to prove the LASGTP and anode are still working after the in situ biasing operation. It's better to show the in situ biasing charge-discharge curve to clarify this point.

Response: We thank the reviewer for pointing out this issue. Figure 1b shows the *in situ* biasing charge-discharge curve of the thinned TEM sample operated in the transmission electron microscope, which shows that the *in situ* anode worked well during the *operando* TEM observation. Moreover, comparison of the charge and discharge curves between the original and FIB processed thin film battery are shown below, and they are also shown in the Supplementary Information of our previous study (DOI: 10.1021/acs.nanolett.8b02587). We confirmed that the electrochemical properties did not change by FIB

processing. Thus, we believe that the FIB processed TEM sample worked properly. Furthermore, the thinned region in the present TEM sample was only the cathode side, as shown in Fig. 1a. This also showed that the LASGTP and anode were still working after the FIB process. We have revised the figure legend of Fig. 1b.

Charge and discharge curves of (a) the original thin film battery and (b) the FIB processed thin film battery. Note that the size of each sample is different (DOI: 10.1021/acs.nanolett.8b02587).

Comment 2: It is necessary to describe the TEM holder and the way to bias the microbattery (STM holder or MEMS chip holder). The previous report had no such information (DOI: 10.1021/acs.nanolett.8b02587). The specific experimental conditions are essential to understand this operando biasing TEM studies.

Response: We thank the reviewer for the suggestion. The configuration of the biasing TEM holder has been reported in our previous study (DOI: 10.1016/j.ultramic.2016.11.019). A schematic of the micro-battery on our biasing TEM holder is shown below. The sample preparation procedure was as follows. First, Cu electrode wires were connected to both sides of the micro-battery using silver paste. Only the cathode side of the battery was then thinned using 30 and 8 kV focused Ga-ion beams. Finally, the thinned TEM sample with the Cu electrode wires was placed on the two biasing electrodes of the TEM holder. As recommended by the reviewer, we have added the configuration of the TEM holder and the method to bias the micro-battery to the Supplementary Information.

Figure S1. Configuration of the biasing TEM holder and the method to bias the micro-battery. The electrode wires and micro-battery were connected using silver paste. Only the cathode side of the micro-battery was thinned by focused-ion beams.

Comment 3: FIB may cause the thickness variance for different parts of the LiCoO₂ cathode especially at the vicinity of boundaries. The influence of thickness on EELS is well-known (DOI:10.1088/0034-4885/72/1/016502). Does it have an influence on the EELS data, especially the sparse coding reconstruction process?

Response: We thank the reviewer for the valuable comment. The thickness variance does not have a large effect on the Li concentration maps measured by the S_A/S_B method because the method measures the concentration through the atomic ratio of Li/Co, where we assume that the Co concentration does not change during the electrochemical reaction. In the paper mentioned by the reviewer (DOI:10.1088/0034-4885/72/1/016502), the following is given on page 18: “If the ratio of two core-loss signals is displayed, the result is an *elemental-ratio* image in which variations in specimen orientation, thickness and beam current are suppressed, according to equation (31).” Therefore, we can also say that the thickness variation does not have a large effect on the Li concentration maps and the sparse coding process. We have added the description to page 6 of the marked copy of the revised manuscript.

Comment 4: On page 14, the calculation method of the average Li concentration is better to be described in detail at the supporting information.

Response: We thank the reviewer for the valuable suggestion. The average Li concentration was calculated as the average value of SC-reconstructed Li maps in the entire cathode film. As recommended by the reviewer, we have added the calculation method of the average Li concentration to the manuscript.

Comment 5: The reason why Li ions were non-monotonic extracted during the charging process should be reasonably discussed (Stage B) since this is very unusual to battery research. This abnormal phenomenon was caused by characterization error or lateral diffusion of Li ions?

Response: As the reviewer mentioned, non-monotonic extraction is a surprising phenomenon in battery research fields. However, we consider that the phenomenon was not caused by characterization error but by the lateral diffusion of Li ions, as the reviewer mentioned, because of the following experimental evidence. One of the reasons is that the average Li concentration in the whole LiCoO₂ film monotonically decreased, as shown in Fig. 5b, which is consistent with galvanostatic charging. This shows that our STEM-EELS correctly measured at least the average Li-ion concentration. Moreover, non-monotonic extraction was suggested in the raw STEM-EELS data without spectrum fitting and sparse coding, as shown in the black plots in Fig. 4f and g. The black plots were calculated from the unprocessed (original) EELS spectra with only spatial averaging. The plots showed noisy but clear non-monotonic extraction, where the changes in the Li concentration were similar to those in Fig. 5c and d. Therefore, we concluded that the Li ions moved in the lateral direction between the LiCoO₂ domains and were non-monotonically extracted in nanoscale local regions. To obtain this new finding, *operando* STEM-EELS with high temporal resolution is very useful. We have added the above description to pages 13–14 of the marked copy of the revised manuscript.

REVIEWERS' COMMENTS:

Reviewer #2 (Remarks to the Author):

In this revised version, the authors have addressed my concerns. I have no more comments. I support publication of this work.

Reviewer #3 (Remarks to the Author):

The authors have addressed my previous concerns, I have no further comment.